# Does the Rural Land Transfer Promote the Non-Grain Production of Cultivated Land in China?

**Yuanyuan Chen** [1], **Mu Li** [1,*] **and Zemin Zhang** [2]

1 School of Public Management, Tianjin University of Commerce, Tianjin 300134, China; chenyy123@tjcu.edu.cn
2 Institute of Ecology, Chinese Research Academy of Environmental Sciences, Beijing 100012, China
* Correspondence: tjculm@tjcu.edu.cn

**Abstract:** In facing the situation of food security, the issue of the non-grain production of cultivated land (NGPOCL) in China has attracted more and more attention. To clarify whether rural land transfer promotes NGPOCL in China, this study collected provincial panel data from 2015 to 2020, and constructed multiple econometric models to explore the impact of land transfer on the planting structure of cultivated land. It is observed that an increase in land transfer area does not promote but significantly inhibits NGPOCL at the national level. The research conclusion is still valid after the robustness test of replacing the explained and core explanatory variables and solving the endogenous problems. The heterogeneity analysis suggests that the inhibitory effect is more pronounced in areas with better topography, economy, or grain production conditions. The analysis of the moderating effect shows that the diversification of land transfer modes and directions can mitigate this inhibitory effect, while the signing of land transfer contracts does not show a significant regulatory effect. This paper reveals the effect of land transfer on grain cultivation from a macro perspective. Its conclusions may provide policy implications for the optimization of rural land transfer and curbing NGPOCL in China.

**Keywords:** food security; rural land transfer; non-grain production; cultivated land

## 1. Introduction

With the impact of the COVID-19 pandemic, economic globalization has encountered a countercurrent and unilateralism is on the rise [1]. International trade has been affected by the closures of ports and the global food supply chain has been disrupted by export bans [2]. In many countries, people's ability to access adequate and nutritious food has been affected, and hunger levels have risen [3]. In facing these new situations, it is of great practical and strategic significance to constantly improve the comprehensive grain production capacity and effectively ensure national food security [4].

Cultivated land is fundamental to grain production [5]. The utilization mode of cultivated land directly determines the capacity of grain production and profoundly affects the national food security pattern [6,7]. The cultivated land in China is limited and decreasing [8]; thus, improving the utilization efficiency and intensity of cultivated land has become the key to ensuring national food security [9,10]. However, the property rights of cultivated land in China are seriously fragmented, which greatly hinders the improvement of its utilization efficiency and intensity [11,12]. Aware of this, the government has gradually enhanced its policy support and guidance for rural land transfer [13]. The orderly transfer of rural land is conducive to cracking the fragmentation of cultivated land, expanding the land operation scale and improving the level of agricultural modernization and food security [14–17]. In 2014, the Chinese government put forward the policy of "separation of three rights" for rural land [18], emblematizing that rural land transfer in China has entered a mature period of policy [19].

With the guidance of the policy, the rural land transfer area in China has been increasing. In 2020, the acreage of transferred cultivated land reached 37.64 million hectares,

accounting for 36.15% of the total cultivated land. However, in many areas, and even in the major grain producing areas, the phenomenon of the non-grain production of cultivated land (NGPOCL) has intensified after the transfer of land [20,21]. NGPOCL directly leads to a reduction in grain planting area, resulting in a decline of grain production. Furthermore, the soil tillage layer and corresponding supporting facilities can be destroyed by some non-grain production types, such as pond fish farming and nursery plantation [22,23]. The crop yield will be influenced even if the planting of grain crops is resumed [24]. As a result, NGPOCL poses a new threat to China's food security [25].

Many scholars have believed that rural land transfer promotes NGPOCL. The main logic is that operators will pay rent as the land is transferred in, which increases the operating costs, so they prefer to grow more profitable cash crops to avoid losing profits [26–29]. In addition, with the land transfer-in and the expansion of the operation scale, the commercial attributes of land products and their economic benefits have become the principal pursuit of operators [27]. As a result, the willingness to obtain more profits gives operators sufficient incentive to increase the cultivation of non-grain crops and reduce the planting of grain crops [30].

At the same time, some scholars have argued that land transfer did not definitely result in NGPOCL, or even in the grain-oriented production of cultivated land (GOPOCL). For example, based on field survey data of 1120 households in the Hubei Province, Peng et al. found that land transfer could significantly increase farmers' tendency to grow grain crops. That is, it has a positive effect on the grain orientation of the crop planting structure [31]. Using the CLDS data, Jiang and Hu believed that rural land transfer and the introduction of industrial and commercial capital to the countryside, in order to provide productive services, are conducive to the expansion of grain production [32]. In addition, the impact of land transfer on the planting area of different grain crops is also different. Liu et al. found that farmers' propensity to grow wheat did not change significantly after the land transfer-in, but that their tendency to plant maize increased in five counties within Gansu Province [33]. Qiu et al. showed that land transfer-in had a positive impact on rice planting areas, especially in the case of the less available labor force for agriculture, based on the survey data of farmers in nine provinces of China [34]. Qian et al. found that farmer's land transfer-in had no effect on the cultivation of wheat and maize, but could significantly expand rice cultivation, based on a systematic analysis of the national survey data of rural fixed observation points in China [35].

In addition, some studies have shown that the relationship between rural land transfer area and crop planting structure is non-linear or will change with the changes of other factors. Through an interview survey of a certain number of farmers located in several provinces in China, Peng et al. found that the area of land transfer-in had a U-shaped impact on the probability of planting grain crops [36]. Based household survey data, Bi et al. [37] and Li and Yang [38] found that there is a threshold effect on the impact of land transfer and the operation scale on planting structure. When the operation scale is less than the threshold scale, the land transfer-in and the expansion of the operation scale will promote the farmers to grow cash crops, but this effect will be reversed when the operation scale exceeds the threshold scale. Luo et al. and Liu et al. believed that small-scale farmers are more inclined to increase the planting scale of cash crops. However, with the migration of agricultural labor and the development of the agricultural service market [39], or the improvement of mechanization operation level [40], the crop planting structure will shift to grain-oriented.

Combing the existing literature, we can find that most of these studies are based on the survey data of households in one or several provinces, carried out by a certain institution or by the authors, and that the type of data and the choice of indicators vary greatly. As a result, these research conclusions are mostly regional and subject-specific, rather than national and universal. Therefore, it is still unclear whether rural land transfer promotes NGPOCL at the national level, and it is necessary to use the full sample data for the analysis of NGPOCL from a macro perspective [25]. Additionally, the quality of the land transfer

in China has also undergone great changes in recent years. The proportion of the land subcontracted area has declined, and more and more cultivated land has been transferred to new types of agricultural operating entities. Furthermore, the proportion of contracted land transfer area increased from 53.2% to 65.6% from 2009 to 2019. These changes may reinforce or mitigate the effect of the land transfer area on NGPOCL [41]. However, few studies have paid attention to this issue, which is not conducive to a comprehensive understanding of the impact mechanism of rural land transfer on the planting structure of cultivated land.

In view of this, first of all, we took each province as an observation object and collected the data related to cultivated land transfer and agricultural production from 2015 to 2020. Then, we constructed multiple regression models to clarify the impact of the land transfer area on NGPOCL and its spatial heterogeneity. Furthermore, we constructed moderating effect models to reveal the modulating effect of the land transfer quality on the impact. The rest of this article is arranged below: Section 2 describes the variable selection, model building, data sources, and descriptive statistics. Section 3 presents the findings of this paper, and some discussion is conducted in Section 4. Section 5 shows the main conclusions.

## 2. Materials and Methods

### 2.1. Variable Selection

#### 2.1.1. Explained Variable

For the purposes of this study, the non-grain production rate (NGPR) of cultivated land was set as the explained variable. This is calculated as the proportion of the planting area of non-grain crops to the total planting area of agricultural crops, which is adopted by most scholars [26,30,42,43].

#### 2.1.2. Core Explanatory Variable

In this study, we focused on the impact of changes in the rural land transfer area on theNGPOCL. Thus, the rural land transfer area (Area) is the core explanatory variable, which is defined by the transfer area of the household contracted cultivated land.

#### 2.1.3. Moderating Variables

In order to investigate whether the quality of rural land transfer reinforces or mitigates the impact of the rural land transfer area on NGPOCL, the diversification of the land transfer mode (Mode) and direction (Direction), and the proportion of land transfer contracts signed (Contract) were selected as the moderating variables. We chose these three variables for two reasons. Firstly, these three variables are often used to reflect the quality of the land transfer. Secondly, the changes of these three variables may have an impact on the possibility of NGPOCL. Theoretically, for a certain region, the more diversified the land transfer mode and direction, the more developed the land transfer market, the higher the transfer rent, and the greater the possibility of non-grain production after land transfer. The higher the proportion of signed land transfer contracts, the lower the possibility of non-grain production after the transfer of cultivated land. The Mode is defined as the proportion of the area of land transferred in ways other than subcontracting (leasing) to the total rural land transfer area. The Direction is calculated by the proportion of the area of land transferred into new types of agricultural operating entities (including the rural cooperatives and agricultural industrialized leading enterprises in this paper) to the total rural land transfer area. The Contract is measured by the proportion of the area of transferred land with signed contracts to the total rural land transfer area.

#### 2.1.4. Control Variables

As a direction of the planting structure adjustment, NGPOCL is fundamentally a result of an operator's pursuit of maximizing returns under established constraints [27]. It is strongly influenced by the characteristics of rural households, the conditions of agricultural production, and the cost–benefits of grain production [23,44,45]. In order to control the changes of these factors, we selected nine variables by referring to the existing

literature [42,46,47] and considering the importance and data availability of the variables. The nine variables are: the quantity of the agricultural labor force (Labor), the employment structure of the labor force (Employment), the possession of cultivated land resources (Land), the mechanization level (Machine), the amount of chemical fertilizer input (Fertilizer), the land transfer rent (Rent), the grain production cost (Cost), the grain production profit (Profit), and the income of rural residents (Income), respectively. The variables of Labor and Employment can reflect the characteristics of rural households, the variables of Land, Machine, and Fertilizer can reflect the conditions of agricultural production, and the variables of Rent, Cost, Profit, and Income can reflect the cost–benefits of grain production. The definition of each variable is shown in Table 1.

**Table 1.** Definition of each variable.

| Variables | Definition | Unit |
|---|---|---|
| NGPR | Proportion of the planting area of non-grain crops to the total planting area of agricultural crops | % |
| Area | Transfer area of household contracted cultivated land | Million hectare |
| Mode | Proportion of the area of land transferred in ways other than subcontracting (leasing) to the total rural land transfer area | % |
| Direction | Proportion of the area of land transferred into new types of agricultural operating entities to the total rural land transfer area | % |
| Contract | Proportion of the area of transferred land with signed contracts to the total rural land transfer area | % |
| Labor | Number of agricultural labor force per household | Person |
| Employment | Proportion of labor force engaged in household operation to total labor force | % |
| Land | Area of cultivated land per household | Hectare |
| Machine | Power of agricultural machinery per hectare | Kilowatt |
| Fertilizer | Input amount of chemical fertilizer per hectare | Kilogram |
| Rent | Land rent per hectare of grain crops * | Thousand yuan |
| Cost | Production cost per hectare of grain crops * | Thousand yuan |
| Profit | Net profit per hectare of grain crops * | Thousand yuan |
| Income | Per capita disposable income of rural residents | Thousand yuan |

Note: * average value of rice, wheat, and maize.

*2.2. Model Building*

2.2.1. Benchmark Regression Model

Based on the above settings for the explained variables, core explanatory variables, and control variables, the following regression model was constructed:

$$NGPR_{it} = \beta_0 + \beta_1 Area_{it} + \sum \beta_k Control_{it}^k + \theta_i + \mu_t + \varepsilon_{it} \tag{1}$$

In the model, $NGPR_{it}$ is the proportion of the planting area of non-grain crops to the total planting area of agricultural crops in the province $i$ in year $t$. $Area_{it}$ is the land transfer area in the province $i$ in year $t$. $Control_{it}^k$ represents the control variable $k$ of the province $i$ in year $t$. $\beta_0$ is the constant term. $\beta_1$ is the coefficient of the land transfer area. $\beta_k$ is the coefficient of the control variable $k$. $\theta_i$ is the provincial dummy variable. $\mu_t$ is the year dummy variable. $\varepsilon_{it}$ is the random disturbance term.

The NGPOCL in China has a large spatial heterogeneity [47,48]. In order to clarify the regional differences of the impact of land transfer on NGPOCL, we constructed regression models for the plain and mountainous areas, eastern, central, and western China, the major grain-producing areas, and the non-major grain-producing areas, respectively. The form of the model is consistent with that of Formula (1).

### 2.2.2. Moderating Effect Model

Referring to the study of Wen et al. [49], the following measurement model was constructed:

$$NGPR_{it} = \beta_0 + \beta_1 Area_{it} + \beta_j Mo_{it}^j + \beta_n Area_{it} \times Mo_{it}^j + \sum \beta_k Control_{it}^k + \theta_i + \mu_t + \varepsilon_{it} \quad (2)$$

In the formula, $Mo_{it}^j$ represents the moderating variable $j$ ($j$ = 1, 2, 3). $Area_{it} \times Mo_{it}^j$ is the interaction between the land transfer area and each moderating variable, and $\beta_n$ is the corresponding coefficient. If $\beta_n$ is significant, the moderating effect is significant. If $\beta_n$ has the same sign as $\beta_1$ (either positive or negative), it means that the effect of land transfer area on NGPOCL is reinforced, and the opposite sign means that the effect is reduced. The other parameters have the same meaning as the Formula (1).

### 2.3. Data Sources and Descriptive Statistics

### 2.3.1. Data Sources

There are multiple data sources for this paper. The data related to agricultural inputs and production in each province were collected from the China Rural Statistical Yearbook [50]. The data related to farmers' management and land transfer in various provinces were derived from the Annual Statistical Report of China's Rural Operation and the Management [51], and the Annual Statistical Report of China's Rural Policy and Reform [52]. The cost and benefit data of three major grain crops were derived from the National Compilation of Costs and Benefits of Agricultural Products [53]. It should be noted that, due to the unavailability of data, the observation objects in this paper do not include Hong Kong, Macao, and Taiwan. Furthermore, the values of some of the indicators in certain provinces are missing, such as those related to the land transfer in Tibet. However, the absence of these data does not affect the estimation of the model. The data test with the Stata 15.1 software shows that the panel data in this paper are still strongly balanced. The research period in this paper is from 2015 to 2020. On the one hand, since 2015, China's rural land system reform has entered a comprehensive deepening stage [14], and rural land transfer has entered a mature period of policy. On the other hand, we consider the availability of provincial data and the consistency of statistical caliber.

### 2.3.2. Descriptive Statistical Analysis

During 2015–2020, the average NGPR of the cultivated land in China was 34.29% (Table 2), 4.87% higher than that in the previous six years. This reflects the fact that the NGPOCL phenomenon in China is intensifying. The mean and median of the NGPR have little difference, meaning that the NGPR roughly shows a normal distribution. There is a big difference between the minimum value and maximum value of the NGPR, implying that the NGPR varies greatly in different provinces. The average annual land transfer area in China has reached 1.15 million hectares, reflecting the situation that land transfer in China has become common. The mean value of the Area is higher than the median, indicating that the Area is right-biased. Like the NGPR, the Area also has significant regional differences, which means that there may be significant regional heterogeneity in the influence of the rural land transfer area on NGPOCL. Detailed statistical descriptions of other indicators are shown in Table 2.

**Table 2.** Basic statistical description of variables.

| Variables | Number | Mean | Median | Std. Dev. | Minimum | Maximum |
|---|---|---|---|---|---|---|
| NGPR | 186 | 34.29 | 34.56 | 15.15 | 2.920 | 64.49 |
| Area | 180 | 1.15 | 0.89 | 0.98 | 0.02 | 4.60 |
| Mode | 180 | 79.84 | 81.56 | 13.29 | 22.92 | 99.44 |
| Direction | 180 | 53.22 | 52.83 | 13.67 | 13.40 | 83.70 |
| Regulation | 150 | 65.33 | 68.72 | 15.55 | 22.14 | 100 |

**Table 2.** *Cont.*

| Variables | Number | Mean | Median | Std. Dev. | Minimum | Maximum |
|---|---|---|---|---|---|---|
| Labor | 182 | 0.78 | 0.77 | 0.28 | 0.06 | 1.44 |
| Employment | 182 | 53.44 | 54.34 | 11.17 | 17.19 | 85.63 |
| Land | 182 | 0.50 | 0.33 | 0.44 | 0.11 | 1.95 |
| Machine | 186 | 6.99 | 5.81 | 3.61 | 3.33 | 26.98 |
| Fertilizer | 186 | 367.8 | 351.9 | 148.0 | 95.40 | 799.6 |
| Rent | 150 | 0.51 | 0.28 | 0.64 | 0 | 2.98 |
| Cost | 156 | 14.67 | 14.08 | 3.03 | 5.25 | 22.63 |
| Profit | 156 | −0.54 | −0.29 | 2.64 | −10.23 | 4.95 |
| Income | 186 | 14.75 | 13.35 | 5.38 | 6.94 | 34.91 |

Table 3 reports the statistics for the NGPR and Area by year and region. It can be noted that the NGPR of cultivated land and the rural land transfer area in different regions of China have shown an increasing trend from 2015 to 2020. Through the horizontal comparison of different regions in the same period, it can be found that the NGPR of the cultivated land in eastern and western China is similar and obviously higher than that in central China. The area of rural land transfer in eastern and western China is almost the same, but both are significantly lower than that in central China.

**Table 3.** Trend changes of NGPR and area by year and region.

| Year | NGPR | | | Area | | |
|---|---|---|---|---|---|---|
| | Eastern | Central | Western | Eastern | Central | Western |
| 2015 | 36.32 | 22.88 | 36.89 | 0.75 | 1.71 | 0.70 |
| 2016 | 36.47 | 21.99 | 37.99 | 0.81 | 1.79 | 0.80 |
| 2017 | 37.45 | 22.27 | 38.63 | 0.88 | 1.90 | 0.84 |
| 2018 | 37.72 | 22.45 | 39.63 | 0.91 | 2.00 | 0.90 |
| 2019 | 38.09 | 23.06 | 40.58 | 0.93 | 2.03 | 0.96 |
| 2020 | 38.39 | 23.17 | 40.82 | 0.98 | 2.03 | 0.96 |

## 3. Results

### 3.1. The Impact of Land Transfer Area on the NGPOCL

#### 3.1.1. Regression Results

Based on the constructed regression model, we adopted the Stata 15.1 software to analyze the impact of the rural land transfer area on NGPOCL. The simulation results of the regress models are presented in Table 4. Model 1 is a regression model with only the core explanatory variables. The control variables for the farmer characteristics are added to Model 2. The control variables for the agricultural production conditions are further added to Model 3. Model 4 contains all the control variables. It can be discovered that the coefficients of the land transfer area in these four models are very close, and all of them are significantly negative at the 0.01 level. That is, with an increase in the rural land transfer area, the proportion of the non-grain crop planting area to the total crop planting area will decline. According to the estimation results of Model 4, for every 1 million hectares increase in the land transfer area, the non-grain rate of the cultivated land will decrease by about 2.85% if the other factors remain unchanged. This proves that the increase in the rural land transfer area does not promote but significantly inhibits NGPOCL in China.

**Table 4.** Simulation results of the regression models.

| Variables | Model 1 | Model 2 | Model 3 | Model 4 |
|---|---|---|---|---|
| Area | −3.2429 *** | −3.2697 *** | −3.5258 *** | −2.8500 *** |
| | (−3.17) | (−3.21) | (−3.46) | (−3.11) |
| Labor | − | −0.2221 | −1.6070 | −0.0272 |
| | − | (−0.07) | (−0.51) | (−0.01) |
| Employment | − | −0.1234 ** | −0.1271 ** | −0.0257 |
| | − | (−2.22) | (−2.29) | (−0.40) |
| Land | − | − | 4.5952 ** | 1.0874 |
| | − | − | (2.01) | (0.48) |
| Machine | − | − | 0.2673 | −0.0319 |
| | − | − | (1.50) | (−0.19) |
| Fertilizer | − | − | −0.0020 | −0.0312 *** |
| | − | − | (−0.28) | (−3.85) |
| Income | − | − | − | −1.0288 *** |
| | − | − | − | (−3.80) |
| Rent | − | − | − | 1.4384 * |
| | − | − | − | (1.68) |
| Cost | − | − | − | −0.0623 |
| | − | − | − | (−0.70) |
| Profit | − | − | − | 0.0601 |
| | − | − | − | (0.54) |
| Cons | 36.6371 *** | 43.6364 *** | 41.9482 *** | 61.4349 *** |
| | (35.12) | (12.89) | (9.59) | (10.19) |
| N | 180 | 180 | 180 | 150 |
| $R^2$ | 0.311 | 0.338 | 0.367 | 0.439 |

Note: * $p < 0.1$, ** $p < 0.05$, and *** $p < 0.01$. The decrease in the number of observations in Model 4 is due to the lack of data on grain cost and benefit indicators in Beijing, Tianjin, Shanghai, Tibet, and Qinghai.

3.1.2. Robustness Test of the Model

In order to verify the validity and robustness of the estimation results, a variety of robustness tests were carried out in the following ways.

- Replace the explained variable:

Referring to the study of Luo et al. [39], we adopted the proportion of the area that was used for planting non-grain crops in the transferred cultivated land to the total area of the transferred cultivated land, to reflect the explained variable NGPR. The regression result shows that the coefficient of the Area is significantly negative at the level of 0.01 (Table 5), indicating that the land transfer area increase inhibits NGPOCL, and that the previous conclusion is robust.

- Replace the core explanatory variables:

With the increase of the land transfer area, the farmers' operation scale is also expanded, and the number of farmers operating on a large scale will, accordingly, increase. Based on this, the number of farmers operating with a cultivated land area of more than 3.33 hectares (Largescale) was used as a proxy variable for the land transfer area. The simulation result shows that the coefficient of the Largescale is significantly negative at the level of 0.01 (Table 5). This implies that the expansion of the operation scale restrains NGPOCL, which is consistent with the previous conclusion.

- Endogenous problem management:

The main endogenous problem that is considered in this paper is reverse causality, as the crop planting structure in a region may affect the progress of land transfer [54]. For example, more land transfer may occur in areas where more grain crops are planted [54]. In this paper, two methods were adopted to deal with this problem. The first method is to lag all the explanatory variables by one period, referring to the research of Li et al. [55]. The second method is the instrumental variable method. Drawing on the ideas of Luo et al. [39],

the number of land transfer arbitration committees was selected as an instrumental variable. The instrumental variable satisfies the two hypotheses of correlation and exogenous. On the one hand, the land transfer area is closely related to the number of land transfer arbitration committees. On the other hand, the crop planting structure in a region is not directly affected by the number of land transfer arbitration committees. The two-stage least squares (2SLS) estimation shows that the estimated coefficients of the instrumental variables in the first stage are significant at the level of 0.01, and that the F value of the weak instrumental variable test is larger than 10, which indicates that there is no weak instrumental variable problem, and that the instrumental variable is valid [56]. The estimated results of the two methods show that the coefficients of the land transfer area are significantly negative at the 0.05 and 0.01 levels, respectively (Table 5). This demonstrates that, after considering the possible endogenous problems, the change in the land transfer area still significantly inhibits NGPOCL. This conclusion is consistent with the previous one, which once again verifies the robustness of the estimated results.

**Table 5.** Results of robustness test.

| Variables | Replace the Explained Variable | Replace the Core Explanatory Variables | Lag Explanatory Variables by One Period | Instrumental Variable Method |
|---|---|---|---|---|
| Area | −7.6759 *** | – | – | – |
|  | (−2.67) | – | – | – |
| Largescale | – | −0.1803 *** | – | – |
|  | – | (−3.26) | – | – |
| L.Area | – | – | −1.6251 ** | – |
|  | – | – | (−2.07) | – |
| Area | – | – | – | −10.9469 *** |
|  | – | – | – | (−6.27) |
| Control Variables | Yes | Yes | Yes | Yes |
| Province | Yes | Yes | Yes | Yes |
| Year | Yes | Yes | Yes | Yes |
| $N$ | 150 | 150 | 125 | 150 |
| $R^2$ | 0.206 | 0.443 | 0.572 | 0.722 |

Note: ** $p < 0.05$, and *** $p < 0.01$.

### 3.2. Spatial Heterogeneity of the Impact

Table 6 displays the model estimation results of different topographic areas, economic belts, and grain production areas. It can be found that the estimation results varied greatly. The coefficient of the rural land transfer area in plain areas is significantly negative at the level of 0.01, while that in mountainous areas is not significant. This implies that the increase in the land transfer area in the plain areas promotes grain cultivation, but the land transfer in the mountains does not affect the crop planting structure. That is, the land transfer in areas with better topographic conditions facilitates grain cultivation. The coefficients of the land transfer areas in eastern and central China are significantly negative at the level of 0.01 and 0.05, respectively. This indicates that the land transfer in these two regions inhibits NGPOCL, and that the inhibitory effect in eastern China is more significant. However, the coefficient of the land transfer area is not significant in western China, meaning that the rural land transfer in this region does not affect NGPOCL. This means that the improvement of the economic level can strengthen the inhibitory effect of rural land transfer on NGPOCL. The coefficient of the land transfer area in the main grain-producing areas is significantly negative at the 0.1 level, while it is not significant in the non-major grain-producing areas. This indicates that the better the grain production conditions, the stronger the inhibitory effect of the land transfer on NGPOCL, and the greater the possibility of planting grain crops after land transfer.

**Table 6.** The estimation results of different topographic areas, economic belts, and grain production areas.

| Variables | Plains | Mountains | Eastern | Central | Western | Main Grain-Producing Areas | Non-Major Grain-Producing Areas |
|---|---|---|---|---|---|---|---|
| Area | −4.8685 *** (−4.58) | 1.2918 (0.97) | −2.9785 *** (−3.75) | −5.4807 ** (−2.10) | 0.4086 (0.16) | −2.0656 * (−1.88) | 0.6209 (0.29) |
| Control Variables | Yes | Yes | Yes | Yes | Yes | Yes | Yes |
| Province | Yes | Yes | Yes | Yes | Yes | Yes | Yes |
| Year | Yes | Yes | Yes | Yes | Yes | Yes | Yes |
| $N$ | 72 | 78 | 48 | 48 | 54 | 78 | 72 |
| $R^2$ | 0.654 | 0.737 | 0.657 | 0.453 | 0.744 | 0.339 | 0.740 |

Note: * $p < 0.1$, ** $p < 0.05$, and *** $p < 0.01$. The extent of plains and mountains refers to Wang et al. [57], the plains include Beijing, Tianjin, Hebei, Shanxi, Inner Mongolia, Liaoning, Jilin, Heilongjiang, Shanghai, Jiangsu, Anhui, Shandong, Henan, Ningxia, Xinjiang, and Tibet, while other regions are mountainous areas. According to China's economic census bulletin, the eastern region includes Beijing, Tianjin, Hebei, Liaoning, Shanghai, Jiangsu, Zhejiang, Fujian, Shandong, Guangdong, and Hainan, the central region includes Shanxi, Jilin, Heilongjiang, Anhui, Jiangxi, Henan, Hubei, and Hunan, and the western region includes Inner Mongolia, Guangxi, Chongqing, Sichuan, Guizhou, Yunnan, Tibet, Shaanxi, Gansu, Qinghai, Ningxia, and Xinjiang. According to the statistical standards of the National Bureau of Statistics, the major grain-producing areas include Heilongjiang, Henan, Shandong, Sichuan, Jiangsu, Hebei, Jilin, Anhui, Hunan, Hubei, Inner Mongolia, Jiangxi, and Liaoning, while other regions are non-major grain-producing areas.

### 3.3. Moderating Effect of the Rural Land Transfer Quality

Table 7 shows the model estimation results after adding the moderating variables. The regression coefficients of "Area × Mode" and "Area × Direction" are significantly positive at the level of 0.05, implying that the diversification of the land transfer mode and direction can mitigate the inhibitory effect of the land transfer area on NGPOCL. That is, for the land transfer of a certain area, with an increase of the proportion of land transferred by ways other than subcontracting (leasing), or the proportion of land transferred into new types of agricultural operating entities, the inhibitory effect of the land transfer area on NGPOCL will decrease; thus, the possibility of non-grain production may increase. The coefficient of "Area × Contract" is not significant, indicating that changes in the proportion of the land transfer contracts signed do not affect the relationship between the land transfer area and the NGPOCL. This reflects that the signing of land transfer contracts does not effectively regulate the planting behavior of agricultural operating entities. In addition, it can be observed that the coefficients and significance of the core explanatory variable change little after adding the moderating variable, which once again proves that the previous conclusion is robust.

**Table 7.** Estimation results of moderating effect of land transfer mode, direction, and contract.

| Variables | Land Transfer Mode | Land Transfer Direction | Land Transfer Contract |
|---|---|---|---|
| Area | −3.2752 *** (−3.72) | −3.0183 *** (−3.34) | −2.8366 ** (−2.26) |
| Mode | 0.0772 ** (2.47) | – – | – – |
| Area × Mode | 0.0870 ** (2.74) | – – | – – |
| Direction | – – | 0.0267 (0.96) | – – |
| Area × Direction | – – | 0.0677 ** (2.33) | – – |
| Contract | – – | – – | −0.0182 (−0.62) |

**Table 7.** *Cont.*

| Variables | Land Transfer Mode | Land Transfer Direction | Land Transfer Contract |
|---|---|---|---|
| Area × Contract | – | – | −0.0612 |
| | – | – | (−1.09) |
| Control Variables | Yes | Yes | Yes |
| Province | Yes | Yes | Yes |
| Year | Yes | Yes | Yes |
| $N$ | 150 | 150 | 125 |
| $R^2$ | 0.500 | 0.467 | 0.418 |

Note: ** $p < 0.05$, *** $p < 0.01$.

## 4. Discussion

### 4.1. Compared with Previous Studies

There have been a large number of studies on the relationship between rural land transfer and NGPOCL. Due to different sample selections, scholars have not formed a unified understanding of this issue. In this study, we explored this issue based on provincial panel data. The results show that an increase in the rural land transfer area can inhibit NGPOCL. This view is in agreement with the studies of Liu et al. [33], Qiu et al. [34], Qian et al. [35], and Peng et al. [31]. Furthermore, in order to verify whether there is a U-shaped relationship between the land transfer area and NGPOCL, as considered by Peng et al. [36] and Zhang and Du [43], we adopted the method proposed by Lind and Mehlum [58] for further analyses. The results show that the hypothesis does not pass the significance test. That is, at the macro level, the increase of the land transfer area can steadily inhibit NGPOCL if the other factors remain unchanged.

In addition, we revealed the spatial heterogeneity of the impact of the land transfer area increase on the NGPOCL in China, and explored the moderating effect of the land transfer modes, directions, and contracts on this impact. These works have rarely been studied systematically before, which may be a marginal contribution of this study.

### 4.2. Interpretations of the Results

The interpretations of the inhibitory impact of land transfer on NGPOCL are manifold. First of all, compared with non-grain crops, grain crops are more suitable for large-scale cultivation and facilitate the use of machinery [59]. The entire process of grain production, from sowing, ploughing, loosening, fertilizing, and irrigation, to harvesting, threshing, and drying, is accompanied by the extensive use of machinery [60]. Because of these stronger mechanical substitutions, grain production requires a smaller amount of labor and lower labor costs [34]. In 2020, the number and costs of the labor required for grain production in China were 66.60 days/hectare and 6.19 thousand yuan/hectare, respectively, and only about 2/3 of oilseeds and 1/5 of cotton and tobacco (Table 8). In recent decades, the numbers of the agricultural labor force in China are decreasing, while the total power of agricultural machinery is increasing. As a result, with the increase of the land transfer area and the expansion of the operation scale, farmers are more inclined to grow grain crops [61].

**Table 8.** The numbers and costs of labor required for grain and cash crops in China in 2020.

| Variables | Crops | National | Eastern | Central | Western |
|---|---|---|---|---|---|
| Numbers of labor required (day/hectare) | Grain * | 66.60 | 71.03 | 58.56 | 112.48 |
| | Oilseeds # | 103.05 | 111.67 | 81.83 | 129.71 |
| | Cotton | 328.35 | 293.80 | 303.34 | 363.45 |
| | Tobacco | 322.80 | 332.90 | 295.89 | 427.20 |

**Table 8.** *Cont.*

| Variables | Crops | National | Eastern | Central | Western |
|---|---|---|---|---|---|
| Costs of labor required (thousand yuan/hectare) | Grain * | 6.19 | 6.63 | 5.61 | 10.22 |
| | Oilseeds # | 9.28 | 10.10 | 7.43 | 11.92 |
| | Cotton | 31.07 | 27.92 | 28.75 | 34.29 |
| | Tobacco | 29.04 | 29.85 | 26.68 | 38.35 |

Note: * average value of rice, wheat, and maize, # average value of peanuts and rape.

Secondly, with the proposal of the policy of "separation of three rights" for rural land, the land management rights of operators have become more stable. As a result, operators are more willing to increase their investments and pursue long-term benefits. Compared with non-grain crops, grain crops have longer growth cycles, a lower market risk, less investment requirement, a better price protection policy, and more stable returns [61]. From 2015 to 2020, the variance of the net profit of grain crops was 55.09, while the variance of the net profit of oil crops, cotton, and flue-cured tobacco was 1.9, 4.2, and 2.7 times that of grain crops, respectively. Therefore, with the development of rural land transfer, operators seeking stable benefits are more inclined to grow grain crops.

Thirdly, compared with cash crops, the production of grain crops can be more sub-engineered, so it is easier to outsource their productive services [62]. In recent years, under the guidance of policies, China's agricultural productive service industry has developed rapidly and innovated. Various market-oriented service subjects have competed for development, and a diversified supporting agricultural financial service system has initially taken shape. Statistics show that the numbers of national farmers' professional cooperatives and county-level land transfer service centers increased to 1.89 million and 20.17 thousand in 2018, respectively, and the amount of agriculture-related loans increased by 35.19 trillion yuan in 2019. With the rapid development of the agricultural productive service market, grain crops are more likely to be selected by farmers through the outsourcing of productive services.

This paper also shows that the inhibitory effect of the rural land transfer on NGPOCL is more pronounced in areas with better topographic, economic, or grain production conditions. In these areas, the mechanization level of grain cultivation is higher and land transfer is more conducive to large-scale operations, so the amounts and costs of the labor required are less (Table 8). Additionally, due to better grain production conditions, the net profits of the grain crops in eastern and central China are 99.9 yuan/hectare and 145.3 yuan/hectare higher than that of those in western China in 2020, respectively. Moreover, better economic development and agricultural industrialization levels in these areas are more conducive to the development of agricultural productive services. Statistics show that the number of land transfer service centers in eastern and central China accounted for 72.9% of the -total in 2018, and the amount of agriculture-related loans accounted for 74.2% of the total in 2019. The higher level of agricultural productive services in these areas further highlights the comparative advantage of grain crops. As a result, with the increase of land transfer area and the expansion of operation scales, grain crops have become a rational choice for more agricultural operating entities in areas with better topographic, economic, or grain production conditions.

Furthermore, this paper concludes that the diversification of the land transfer modes and directions can mitigate the impact of the land transfer area on NGPOCL. The main logic lies in the fact that land subcontracting (leasing) mostly occurs among ordinary farmers, while other modes of land transfer (such as shareholding) mainly occur between farmers and rural cooperatives or agricultural enterprises. In general, ordinary farmers grow crops mainly based on planting habits or to meet their own ration needs, while large-scale farmers are more profit-seeking in their agricultural production, and cost–benefit is the basis for their decision making [37]. Therefore, the non-grain production tendency of ordinary farmers is lower than that of the new types of agricultural operating entities [24,27,63], and the higher the proportion of the land transferred into these new types of agricultural

operating entities, the weaker the inhibitory effect of the land transfer area on NGPOCL. The inhibitory effect of the land transfer area on NGPOCL is not affected by the proportion of the cultivated land area that is signed with transfer contracts, which is not in line with policy expectations. This may be due to the land utilization type of the transferred land not being clearly specified in the transfer contract, or the legal effect of the land transfer contract being insufficient.

### 4.3. Discrimination of NGPOCL and GOPOCL

Although this study shows that rural land transfer has a positive impact on grain cultivation, it is undeniable that the transferred cultivated land in many areas has been used for non-grain production. In fact, both the trends of NGPOCL and GOPOCL have their own reasonable formation mechanisms. The lower economic benefits of growing grain, the increase in grain production costs such as land rent, the pursuit of higher profits by farmers, the adjustment of the household consumption structure, the continuous influx of industrial and commercial capital into the countryside, and the one-sided understanding of policies by some governments, are all important driving factors for NGPOCL [24,25,27,28,34,42]. As mentioned above, the decline in agricultural labor, the convenience of mechanized production, and the relatively low risks and advantages of service outsourcing have driven operators to grow grain crops. Whether to grow grain crops or non-grain crops is a rational choice made by various operating entities, based on their comprehensive considerations of "economic rationality" and "survival rationality" [63].

In recent years, with the migration of the agricultural labor force and the adjustment of agricultural policies, the differentiation of rural households in China has rapidly deepened. In addition to ordinary farmers, some farmers have evolved into part-time farmers, some have developed into family farms through land transfer, and some have developed into professional cooperatives or agricultural enterprises through the introduction of industrial and commercial capital. Different operating entities have different resources and requirements, so their willingness and ways of using agricultural land are also quite dissimilar. Generally speaking, the agricultural production activities of ordinary farmers and part-time households are not oriented towards commercialization and profit, and they may grow grain crops just because of their cultivation habits. Family farms have a stronger market awareness and profit-seeking nature, so they have a relatively strong non-grain production tendency. Agricultural enterprises are more likely to pursue returns on their investments, and their willingness to participate in non-grain production is strongest among these operators. Therefore, whether the transferred land is used for non-grain production or grain-oriented production, the type of operators to which the land is transferred may be crucial.

### 4.4. Policy Implications

The findings of this study may provide the following policy implications. Above all, the government (especially in the major grain-producing provinces of eastern and central China) should continue to promote the orderly transfer of rural land. Multiple measures can be taken, such as constructing more rural land transfer service institutions, building more rural land transfer platforms at the county, town, and village levels, increasing the publicity, consultation, and guidance services for rural land transfer, and so on. Through these measures, more farmers who are not experienced with farming, unwilling to farm, and unable to farm will be encouraged to transfer their land to operating entities that are effective at farming, willing to farm, and capable of farming, thereby improving the effectiveness of resource allocation and promoting moderate-scale operations and agricultural modernization.

Furthermore, against the backdrop of dwindling agricultural labor, it is necessary to promote the high-quality development of the agricultural productive service industry. The following measures can be taken: the integration of agricultural productive service industry support policies into the agricultural support and protection policy system; an

increase in the support for agricultural service entities through optimization policies such as subsidies; strengthening the development of the agricultural productive service market; and the establishment of more regional, multi-type, and multi-center trading platforms for various agricultural production services.

In addition, the categorical management of agricultural operating entities should be strengthened. Currently, and for a long time to come, smallholder family operations will continue to be the main mode of agricultural operation in China. It is necessary to attach importance to the role of traditional farmers in ensuring national food security, and to strengthen the policy and technical supports available to them. For new agricultural operating entities, the supervision and control of land use should be reinforced, and their leading roles to small farmers should be enhanced. Furthermore, effective ways to connect smallholder farmers with modern agricultural development should be continuously explored, and a coordinated development pattern between these small farmers and new agricultural operating entities should be constructed.

Lastly, it is necessary to introduce a national standard for land transfer contracts as soon as possible and strengthen their management. The utilization type of the transferred land should be clearly specified in the contract. For example, cultivated land for growing grain crops should only grow grain crops after its transfer, and the cultivated land for growing cash crops should grow grain, cash crops, or fodder crops after its transfer. Additionally, the contract should detail and concretize the responsibilities of and punishments for agricultural operating entities should they violate regulations, and the management department should enhance the supervision of the land utilization of transferred land. This can be considered to establish an early warning system for the non-grain production of transferred cultivated land, with the help of remote sensing technology.

*4.5. Research Limitations and Future Directions*

Due to the unavailability of data, some limitations exist in this paper and need to be further studied. For example, although the results of this paper reflect that the non-grain production tendency of ordinary farmers is lower, the change characteristics of different agricultural operating entities' planting behavior after land transfer-in have not been systematically analyzed. Identifying this problem can provide clearer analytical logic and more targeted policy recommendations for the non-grain production of transferred cultivated land [21], which should be a direction worthy of future research. In addition, scholars have not yet formed a unified understanding of the classification standards and types of agricultural operating entities. Therefore, how to scientifically and reasonably divide and judge these agricultural operating entities may also be a topic worth studying in the future.

Furthermore, many studies have shown that rural land transfer can also greatly affect the yield of grain crops by influencing the efficiency of scientific and technological utilization and management [64–67]. Therefore, rural land transfer can affect the two direct determinants of grain production, and will undoubtedly have a complex influence on the grain production capacity. Under the situation of food security, it is of great practical significance to clarify the impact mechanisms of land transfer on grain production, and to quantitatively evaluate the contribution of land transfer. Therefore, relevant studies should be paid more attention to.

## 5. Conclusions

This paper explored the impact of land transfer on NGPOCL in China based on the provincial panel data. The results showed that an increase in the rural land transfer area significantly inhibits NGPOCL at the national level. The research conclusion was still valid after the robustness tests of replacing the explained and core explanatory variables and solving the endogenous problems. The heterogeneity analysis suggested that the inhibitory effect of rural land transfer on NGPOCL was more pronounced in areas with better topography, economy, or grain production conditions. The analysis of the moderating

effect showed that the diversification of the land transfer modes and directions could mitigate the inhibitory effect, while the signing of land transfer contracts did not show a significant regulatory effect. Based on these findings, some policies and measures should be adopted to optimize rural land transfer and curb NGPOCL.

**Author Contributions:** Conceptualization, Y.C. and M.L.; methodology, Y.C.; software, Y.C.; validation, Y.C., M.L. and Z.Z.; formal analysis, Y.C.; investigation, Y.C.; resources, Y.C.; data curation, Y.C. and M.L.; writing—original draft preparation, Y.C.; writing—review and editing, Y.C.; visualization, Y.C.; supervision, M.L.; project administration, M.L. and Z.Z.; funding acquisition, M.L. and Z.Z. All authors have read and agreed to the published version of the manuscript.

**Funding:** This research was funded by the National Natural Science Foundation of China (Grant number 42101266) and Tianjin Philosophy and Social Science Research Planning Project (Grant number TJGL21-030).

**Data Availability Statement:** The associated dataset of the study is available upon request to the corresponding author.

**Conflicts of Interest:** The authors declare no conflict of interest.

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
