# Peer review of "Does the Rural Land Transfer Promote the Non-Grain Production of Cultivated Land in China?"

_land, doi:10.3390/land12030688_

Round 1

Reviewer 1 Report

It was a pleasure to serve as a reviewer for this paper. The major concern raised in this article is the 'novelty' that needs to be addressed and clarified by the authors.

Point 1. It is recommended that authors include proper citations for the collected data used in their works.

·       Line no. 181 - China Rural Statistical Yearbook

·       Line no. 183 -  Annual Statistical Report of China's Rural Operation and the Management

·       Line no. 184 - Annual Statistical Report of China's Rural Policy and Reform

Point 2. On line 452, there should be written: "Due to the unavailability of data”.

Point 3: This paper is using the majority of the work which has already been published by IJERPH and sustainability journal doi:

https://doi.org/10.3390/ijerph192315587 ,

https://doi.org/10.3390/ijerph192416630 &

https://doi.org/10.3390/su15010379 .

The novelty of the article is questionable; having another similar article will not draw a substantial amount of new readership.

Author Response

Thank you very much for the positive and constructive comments and suggestions on our manuscript. Revised portion are marked in red in the paper and the responds to the comments are as flowing:

Point 1: It is recommended that authors include proper citations for the collected data used in their works. Line no. 181 - China Rural Statistical Yearbook. Line no. 183 -  Annual Statistical Report of China's Rural Operation and the Management. Line no. 184 - Annual Statistical Report of China's Rural Policy and Reform.

Response 1: Thank you for the suggestions. We have added citations of these documents in our manuscript.

Point 2: On line 452, there should be written: "Due to the unavailability of data”.

Response 2: Thanks for pointing out this mistake, we have revised it. 

Point 3: his paper is using the majority of the work which has already been published by IJERPH and sustainability journal doi: https://doi.org/10.3390/ijerph192315587, https://doi.org/10.3390/ijerph192416630 & https://doi.org/10.3390/su15010379. The novelty of the article is questionable; having another similar article will not draw a substantial amount of new readership.

Response 3: Indeed, the research theme of this manuscript and these three papers can be summarized as the impact of land transfer on the non-grain production of cultivated land. However, the perspective, purpose and content of research of this article and the previous three papers are quite different, and the research in this article is still somewhat innovative.  
The majority work of the paper (https://doi.org/10.3390/ijerph192315587) is investigating the impact of rural land transfer on farm households’ non-farm employment. The main conclusion is that the rural land transfer significantly increased the proportion of non-farm employment personnel in farm households and the months of per year non-farm employment per person. Based on this, it is proposed to raise the income of farm households and narrow the gap between urban and rural areas by improve land transfer system. The perspective and focus of its research is the employment and income of farmers. Besides, the research data of this paper is field survey of farmers in Hubei Province in 2018 and the conclusion may be regional and subject-specific.
The object of the paper (https://doi.org/10.3390/ijerph192416630) is to explore the spatial–temporal evolution of non-grain production in thirteen major grain-producing provinces in China and then identify the driving factors. The analysis of the spatial-temporal evolution of non-grain production accounts for a large part of the article, and land transfer is not the main driving factor of cultivated land non-grain according to the research results.
The aim of the paper (https://doi.org/10.3390/su15010379) is to estimate different land leasing entities’ intentions and drivers to grow non-grain crops. This paper revealed the difference in the non-grain use of the transferred farmland from the perspective of the differentiation of the renting entities. It focuses on the impact of operators' location and transaction costs on farmers' planting behavior, rather than the impact of quantity and quality of land transfer on the non-grain production of cultivated land.
The novelty of the article is as follows: Firstly, most of previous studies are based on the survey data of households in one or several provinces carried out by a certain institution or by the authors, the type of data and the choice of indicators are very different, and the research conclusions are mostly local and subject-specific rather than national and universal. In this paper, we took each province as the observation object and used the provincial data for analysis. It avoids variable selection bias and regional limitations, and can reveal the policy effect of China’s rural land transfer at the macro level.
Secondly, we revealed the spatial heterogeneity of the impact of the land transfer area increase on non-grain production of cultivated land in China, which has rarely been studied before because this cannot be done based on local and micro-level farmer survey data.
Thirdly, in addition to the quantity of land transfer, the impact of land transfer quality on the non-grain production of cultivated land is also explored in this paper, which has been less seen in previous studies. In particular, the impact of the signing of land transfer contracts on the non-grain production of cultivated land has obtained unexpected results in this paper, which has strong practical guiding significance.

Reviewer 2 Report

1. In the introduction section, statements such as “The report to the 20th National Congress of the Communist Party of China particularly emphasized that …… indicated the significance of food safety for China, and it seems like a manuscript specifically published for Chinese readers. In fact, this paper is intended for global readers, it is much more important to highlight the significance of global food safety.  

2. Some cases in line 71-94 is too old. Please update the latest cases and data.

3. In variable selection, why variables related to the natural conditions of farmland are not included?

4.Interpretations of the Results should be strengthened.

Author Response

Thank you very much for the positive and constructive comments and suggestions on our manuscript. Revised portion are marked in red in the paper and the responds to the comments are as flowing:

Point 1: In the introduction section, statements such as “The report to the 20th National Congress of the Communist Party of China particularly emphasized that ……” indicated the significance of food safety for China, and it seems like a manuscript specifically published for Chinese readers. In fact, this paper is intended for global readers, it is much more important to highlight the significance of global food safety.  â€¨

Response 1: Thank you for the suggestions. We changed the statements with analyzing the new changes in global food security and highlighting the significance of global food security. Details in line28-38 (revised version with track changes).

Point 2: Some cases in line 71-94 is too old. Please update the latest cases and data.

Response 2: Thanks for the suggestions. We updated the cases with the latest studies of Jiang and Hu (2021), Bi et al (2020) and Li and Yang (2021). After updating, most studies were conducted in the past three years. Details in line75-110 (revised version with track changes).

Point 3: In variable selection, why variables related to the natural conditions of farmland are not included?

Response 3: Thanks for the comments. The variables related to the natural conditions of farmland are not included mainly for two reasons. On the one hand, macro speaking, the natural conditions of farmland in a region are usually unchanged for a long time, and this paper constructs a fixed-effect model of province and year to control the influence of natural conditions of farmland on planting structure. On the other hand, for the variables related to natural conditions of farmland such as soil organic matter content, soil texture, etc., there is a lack of sufficient panel data like other variables, which is also the limitation of this paper. In the future, when conducting micro-farm surveys, the variable related to natural conditions of farmland should be included.

Point 4: .Interpretations of the Results should be strengthened.

Response 4: Thanks for the suggestions. We have strengthened the interpretations of the results. Details in line 365-446 (revised version with track changes).

Author Response

Thank you very much for the positive and constructive comments and suggestions on our manuscript. Revised portion are marked in red in the paper and the responds to the comments are as flowing:

Point 1: The current literature review showed many case studies and the views of different scholars on the effects of the land transfer on NGPOCL, but there are no generalized views or concluding remarks. The current literature review is a bit casual and loose. So, the literature review could be improved.

Response 1: Thank you for the suggestions. We combed the references and revised the literature review. Literature is categorized by point of view, and literature with the same point of view is sorted by year of publication. Details in line 64-114  (revised version with track changes).

Point 2: There is only a general mention of research results in the Compared with Previous Studies Section, which is not sufficient. There is so little summary of the previous studies that we could draw no conclusions on the limitation of previous studies in this section. And how does this manuscript compensate for the limitations?

Response 2: Thank you for the suggestions. We added relevant statements including limitations of previous studies and how this study compensates for the limitations. Details in line 341-348 (revised version with track changes).

Point 3: The Discrimination of NGPOCL and GOPOCL Section is far away from the research topics which focus on the relationship between rural land transfer and non-grain production of cultivated land.

Response 3: Thank you for the suggestions. We quite agree with you and delete this section. And in the Research Limitations and Future Directions Section, we added relevant discussion.

Point 4: What new things can the paper contribute to food security or against hunger in the world? How to link the findings and conclusions in this manuscript with the previous findings and conclusions from other countries? Its introduction, analysis, and discussions should be beyond the local case itself. This can help the manuscript to attract more international readers. However, there are still deficiencies in the contribution of global knowledge.

Response 4: Thank you for the suggestions. We added discussion about the possible contribution of the conclusions of this paper to the food security in other countries. Details in line 517-525 (revised version with track changes).

Round 2

Reviewer 1 Report

The manuscript has been substantially improved by the authors. The new version eliminates any confusion from the previous one. All flaws in the manuscript have been resolved. As it stands, I would like to suggest a minor changes:

Point 1: Tables 4, 5, and 7 might benefit from having blank fields replaced with dashes (-) for better readability.

Author Response

Thank you very much for the positive and constructive comments and suggestions on our manuscript. Revised portion are marked in red in the paper and the responds to the comments are as flowing:

Point 1: Tables 4, 5, and 7 might benefit from having blank fields replaced with dashes (-) for better readability.

Response 1: Thank you for the suggestion. We have followed your suggestion and revised it.

Reviewer 3 Report

The author has made serious and effective modifications to the manuscript. I suggest the authors consider the following comments to improve the manuscript.

1.       In Section 2.1, the selection basis of key variables and control variables in the model needs to be further explained.

2.       The region division in the paper should clearly state the provinces involved in the region division and the basis for the division, such as the eastern, central and western regions in Section 2.3.2, as well as the topographic area, ecological belts and grain production areas in Section 3.2.

3.       The author mentioned in Section 2.3.1 that some indicators of some provinces are missing or wrong. Although this does not affect the overall estimation of the model, will it affect the research of some subregions involved in the paper, such as the statistical results of the eastern, central and western regions in Section 2.3.2? Just like the Tibet region mentioned in the paper, it is an important part of western China. The absence of data from Tibet is critical to the experimental results in western China.

Author Response

Thank you very much for the positive and constructive comments and suggestions on our manuscript. Revised portion are marked in red in the paper and the responds to the comments are as flowing:

Point 1: In Section 2.1, the selection basis of key variables and control variables in the model needs to be further explained.

Response 1: Thank you for the suggestion. We added some explanations about the selection basis of variables. Details in line 144-180 (revised version with track changes).

Point 2: The region division in the paper should clearly state the provinces involved in the region division and the basis for the division, such as the eastern, central and western regions in Section 2.3.2, as well as the topographic area, ecological belts and grain production areas in Section 3.2.

Response 2: Thank you for the suggestion. Referring to the practice of Wang et al., the plains include Beijing, Tianjin, Hebei, Shanxi, Inner Mongolia, Liaoning, Jilin, Heilongjiang, Shanghai, Jiangsu, Anhui, Shandong, Henan, Ningxia, Xinjiang, and Tibet, while other regions are mountainous areas. According to China's economic census bulletin, the eastern region includes Beijing, Tianjin, Hebei, Liaoning, Shanghai, Jiangsu, Zhejiang, Fujian, Shandong, Guangdong and Hainan, the central region includes Shanxi, Jilin, Heilongjiang, Anhui, Jiangxi, Henan, Hubei and Hunan, and the western region includes Inner Mongolia, Guangxi, Chongqing, Sichuan, Guizhou, Yunnan, Tibet, Shaanxi, Gansu, Qinghai, Ningxia and Xinjiang. According to the statistical standards of the National Bureau of Statistics, the major grain producing areas include Heilongjiang, Henan, Shandong, Sichuan, Jiangsu, Hebei, Jilin, Anhui, Hunan, Hubei, Inner Mongolia, Jiangxi and Liaoning, while other regions are non-major grain producing areas. These statements are also added to the article.

Point 3:  The author mentioned in Section 2.3.1 that some indicators of some provinces are missing or wrong. Although this does not affect the overall estimation of the model, will it affect the research of some subregions involved in the paper, such as the statistical results of the eastern, central and western regions in Section 2.3.2? Just like the Tibet region mentioned in the paper, it is an important part of western China. The absence of data from Tibet is critical to the experimental results in western China.

Response 3: Thank you for pointing out this problem. There is no relevant data on land transfer in Tibet in the statistical bulletins over the years, and we cannot effectively estimate it. This may be due to the fact that Tibet's agricultural production is dominated by animal husbandry, with little grain cultivation (the grain planting area in 2020 is only 170.86 thousand hectares), and the internal driving force and objective conditions for land transfer are not sufficient. However, we strongly agree that all data should be included to avoid selective bias. We will address this issue in future studies.
